# The Wrist as a Weightbearing Joint in Adult Handstand Practitioners: A Cross-Sectional Survey of Chronic Pain and Training-Related Factors

**DOI:** 10.3390/jfmk10040372

**Published:** 2025-09-26

**Authors:** Noa Martonovich, David Maman, Assil Mahamid, Liad Alfandari, Eyal Behrbalk

**Affiliations:** 1Department of Orthopedic Surgery, Hillel Yaffe Medical Center, Hadera 3820302, Israel; 2Rappaport Faculty of Medicine, Technion—Israel Institute of Technology, Haifa 3200003, Israel

**Keywords:** overuse injuries, athletic injuries, orthopedics, musculoskeletal pain, exercise, wrist joint

## Abstract

**Background:** Chronic wrist pain is becoming increasingly recognized among athletes engaging in wrist-loading activities such as handstands. However, its prevalence and associated risk factors in handstand practitioners have not been systematically studied. This study aimed to investigate the prevalence of chronic wrist pain and to explore associated factors such as discipline, training habits, and pain management strategies. **Methods:** This cross-sectional study aimed to investigate the prevalence and associated factors of chronic wrist pain among handstand practitioners. Eligible participants were individuals aged 18 years or older, of any gender, who practiced handstands regularly (defined as at least once per week). Participants were recruited via a combination of open invitations on social media (Facebook, WhatsApp, Instagram) and direct outreach to movement studios and training communities. The survey was administered online using Google Forms and remained open for two months. Participation was voluntary and anonymous. Descriptive statistics were used to present sociodemographic characteristics, including age group, gender, sport discipline, and weekly training hours. Participants reported training habits, equipment use, pain history, and management strategies via a self-developed questionnaire designed for this study. Chronic pain was defined as recurring or persistent wrist pain. Descriptive statistics were used to summarize responses. Associations between chronic wrist pain and survey variables were analyzed using Chi-square or Fisher’s exact tests for nominal data, and Chi-square test for trend for ordinal data. A *p*-value < 0.05 was considered statistically significant. **Results:** A total of 321 participants were included in the study. The most represented age group was 25–34 years, comprising 123 (38.3%) of the participants. Gender distribution was 174 (54.2%) males and 147 (45.8%) females. The most common sport disciplines were Yoga (88, 27.4%), Capoeira (60, 18.7%), and Movement (52, 16.2%). Chronic wrist pain was reported by 182 (56.7%) of participants. Younger age was significantly associated with higher pain prevalence (*p* = 0.042). No significant associations were observed between chronic pain and weekly training hours, warm-up routines, brace use, or grip device use. Female participants demonstrated more proactive pain management behaviors (*p* = 0.016). Sport discipline and training practices showed non-significant trends toward pain differences. **Conclusions:** Chronic wrist pain is common among handstand practitioners, particularly among younger athletes. These findings suggest that injury risk may relate more to training intensity and biomechanics than to simple training volume. Further research incorporating objective diagnostics and standardized intervention protocols is warranted.

## 1. Introduction

The human body is biomechanically designed for upright posture, with the lower extremities and axial skeleton bearing the body’s weight during movement and stance. In contrast, the joints of the upper extremities are not anatomically adapted for sustained weight-bearing. Repeated loading through the arms—such as during wheelchair use or crutch walking—can result in various upper limb pathologies, including tendinopathies, ligament or muscle injuries, osteoarthritis, and even fracture [1].

During wrist movements such as flexion and extension, the primary agonist muscles include the extensor carpi radialis (ECR), extensor carpi ulnaris (ECU), and extensor digitorum (ED) for extension, and the flexor carpi radialis (FCR), flexor carpi ulnaris (FCU), and flexor digitorum superficialis (FDS) for flexion [2]. Notably, wrist extensors act as postural stabilizers, often engaging isometrically to counterbalance flexor activity, especially during gripping or load-bearing tasks. Their elevated baseline activation reflects a co-contraction strategy that stabilizes the wrist against perturbations [3]. the finger extensors and flexors not only control digital movement but also contribute to wrist stability due to their cross-joint nature, especially in extended wrist postures [2]. In addition, muscles crossing the elbow joint, such as the biceps and triceps brachii, function as synergists during wrist movement by maintaining upper limb alignment and contributing to overall kinetic chain stability [3,4].

Biomechanically, a static handstand places extreme demands on wrist joint stabilization. To maintain equilibrium, wrist flexors (specifically FCR) must contract isometrically to balance the moment generated by body weight, while wrist extensors resist dorsiflexion, contributing to joint stability [5,6].

During press-to-handstand movements, wrist joint moments increase significantly—reaching approximately 40% of body weight during the transition from toe-off to handstand completion—requiring coordinated activation of both wrist flexors and extensors to manage load and maintain balance [5,7]. During handstand walking, preparatory postural adjustments at the wrist help regulate center-of-pressure shifts, and variability in wrist torque is associated with the dynamic demands of balance control [8]. In acrobatic maneuvers such as round-offs and cartwheels, the wrist absorbs high ground reaction forces during rapid loading and extension, with peak forces exceeding 1.5 times body weight and substantial ulnar deviation torque demands, particularly during contact and push-off phases [9,10]

In recent years, the rise of social media and the popularity of visually impressive physical skills [11,12] have likely contributed to the increasing prevalence of handstand training across diverse disciplines, including yoga, CrossFit, calisthenics, and movement-based practices. During handstands, the wrists function as the primary weight-bearing joint and are often held in extended or hyperextended positions [5]—angles that are atypical in daily life. Despite the growing popularity of handstand practice, the consequences of repetitive wrist loading remain poorly understood. Previous studies have examined musculoskeletal injuries in sports where handstands are frequently performed, including CrossFit, breakdancing, yoga, capoeira, and circus arts and report wrist-related injuries as part of broader injury profiles [13,14,15,16,17,18,19]. In breakdancing, wrist sprains and chronic overuse injuries are prevalent due to repetitive weightbearing and impact during moves such as flares and freezes [13,14]. Yoga practitioners commonly report wrist pain associated with prolonged weightbearing in poses like downward dog and plank, with some case reports describing overuse synovitis or stress reactions [15]. Among circus artists, chronic wrist pain and tendinopathies are common, particularly in disciplines involving aerial and acrobatic work [16]. CrossFit athletes experience wrist sprains, tendinitis, and cartilage injuries from repeated handstand push-ups and bar work [17]. In Capoeira—a martial art characterized by inverted positions, acrobatic transitions, unpredictable movements, and frequent falls onto the ground—the wrist is among the most frequently reported sites of pain, although specific pathologies are often not clearly defined [18]. A case report also described a distal radial stress fracture in a young Capoeira practitioner, highlighting the potential for overuse injuries due to repetitive wrist loading [19].

While these sport-specific studies highlight the presence of wrist-related complaints, they typically report wrist pain as part of a broader injury profile and rarely focus on handstand-specific mechanisms. In fact, most of the available literature addressing wrist injuries has focused on gymnasts [20,21,22]—an athletic population exposed to high-impact, high-volume training from a young age. In this group, wrist pain and injury are highly prevalent, with studies reporting that up to 88% of competitive gymnasts experience wrist pain at some point during their training careers [21]. Common overuse injuries include distal radial epiphysitis (so-called “gymnast’s wrist”), stress fractures of the distal radius, and chronic ligamentous or cartilage injuries due to repetitive axial loading in wrist extension [20,22]. Such pathologies often develop during early adolescence and may impact skeletal development and long-term joint integrity [21]. These injury patterns are largely attributed to repetitive compressive loading, excessive dorsiflexion, and insufficient recovery periods typical of elite gymnastics training [20,22]. While these findings are informative, they may not be generalizable to recreational or adult athletes who participate in lower-impact, handstand-centric disciplines [23,24].

Understanding the prevalence, patterns, and possible training-related risk factors for chronic wrist pain in handstand practitioners is increasingly important, especially as non-elite adults take on wrist-loading activities in yoga and bodyweight sports. While no gold-standard prevention strategy currently exists, gradual load progression, emphasis on wrist mobility, proprioceptive training, and strengthening of stabilizing musculature are commonly proposed to reduce injury risk in wrist-loading sports [25,26,27]

To date, no study has systematically investigated the burden of wrist pain in this emerging population. The current study seeks to address this gap by identifying pain prevalence, associated risk factors, and common management strategies among adult handstand practitioners. This study aims to characterize wrist pain patterns in handstand practitioners and explore associations with training discipline, load, and management strategies.

## 2. Materials and Methods

This cross-sectional study was designed to investigate the prevalence and associated factors of chronic wrist pain among athletes who regularly perform handstands. Participants were recruited through a combination of social media platforms (e.g., Facebook groups, WhatsApp groups, Instagram pages), and direct contact with relevant movement studios and training communities. The survey was administered online via Google Forms and was available for a two-month period. The study was approved by the Institutional Ethics Committee, all participants provided electronic informed consent prior to completing the survey.

This study was conducted in Israel, and although participation was open internationally, all respondents were ultimately Israeli due to recruitment through Hebrew-language channels and local gyms. The questionnaire was originally written and administered in Hebrew and was translated into English after data collection for reporting purposes. Inclusion criteria were age older than 18 years, handstand practice at least once a week, minimum of 6 months of handstand training experience, and provided informed consent. Exclusion Criteria were history of wrist trauma, handstand practice less than once a week, handstand training duration less than 6 months, or incomplete filled responses. Participants who answered “no” to pain and either responded or did not respond to subsequent pain-related questions were still considered to have completed the form correctly.

The questionnaire was written in Hebrew and translated into English for analysis. It consisted of three main sections: 1. Demographics and background: age, gender, and primary sport discipline. 2. Training Habits: years of experience, frequency of training, warm-up routines, use of wrist braces or grip devices (e.g., parallettes, blocks), and other exercises involving wrist hyperextension (such as the planche or press-ups). 3. Wrist Pain and Injury History: presence and description of wrist pain (including onset and duration), pain management strategies, and prior wrist trauma or surgery (e.g., distal radius fracture). The questionnaire was specifically designed for this study and has not been previously validated or published. The full questionnaire and response distribution are provided in Table 1.

For this study, chronic wrist pain was defined as persistent and recurring pain. Other responses were classified as no chronic wrist pain.

Data were exported from Google Forms and processed in Microsoft Excel prior to analysis using IBM SPSS Statistics (version 26.0). Descriptive statistics (absolute frequencies and percentages) were used to summarize demographic and survey responses. Associations between the presence of chronic wrist pain and categorical variables were analyzed. Nominal categorical variables were evaluated using Pearson’s Chi-square test or Fisher’s exact test, depending on expected cell counts. Ordinal categorical variables were analyzed using the Chi-square test for trend (linear-by-linear association).

No formal sample size calculation was performed due to the exploratory nature of the study. Participant inclusion was maximized through open online recruitment over a two-month period. A two-tailed *p*-value of <0.05 was considered statistically significant.

## 3. Results

A total of 434 individuals completed the online questionnaire. Following the application of predefined exclusion criteria, 113 responses were excluded, resulting in a final analytic sample of 321 participants. The participant flowchart (Figure 1) details the exclusion process and final sample selection.

The most represented age group was 26–35 years, accounting for 162 participants (50.4%). Gender distribution included 174 males (54.2%), 147 females (45.8%), and no participants identifying as other. With respect to sports disciplines, yoga was the most common, reported by 88 participants (27.4%). Full demographic details are presented in Table 2.

Chronic wrist pain was reported by 182 participants (56.7%), while 139 (43.3%) reported no chronic wrist pain. Among those without chronic pain, 30 individuals (21.6%) had never experienced wrist pain, whereas 109 (78.4%) reported experiencing it once or twice in the past. The prevalence of chronic wrist pain was highest among participants aged 18–25 (63.8%), followed by those aged 26–35 (57.4%), 36–45 (54.1%), and 46+ (41.4%), although this trend did not reach statistical significance (*p* = 0.223). No significant associations were found between chronic wrist pain and gender (*p* = 0.626), primary sport discipline (*p* = 0.739), or participation in wrist hyperextension exercises such as planche-type training (*p* = 0.818). Similarly, neither grip device use (*p* = 1.000) nor wrist brace use (*p* = 0.418) was associated with a difference in pain prevalence. The duration of handstand training (*p* = 0.758) and weekly training frequency (*p* = 0.455) also showed no significant relationship with the presence of chronic wrist pain. Finally, no significant differences were observed in pain prevalence between participants who performed warm-up routines and those who did not (*p* = 0.171), nor among different types of warm-up routines (*p* = 0.144).

All comparisons between participants with and without chronic wrist pain are summarized in Table 3, Table 4 and Table 5.

Additionally, participants were asked to report their strategies for managing wrist pain. A statistically significant association was found between gender and injury response (*p* = 0.016) as shown in Table 6. In contrast, no significant association was observed between age groups (*p* = 0.520).

## 4. Discussion

This study investigated the prevalence of chronic wrist pain among a diverse cohort of handstand practitioners and explored its associations with demographic and training-related factors. In our sample, over half of the participants (56.7%) reported chronic wrist pain—a prevalence substantially higher than that reported in the general population (4–6%) and even among those engaged in physically demanding occupations or athletic activity (10–24%) [28]. These findings highlight the elevated burden of wrist pain associated with regular handstand practice.

Warm-up practices are traditionally advocated as a preventive strategy across various sports. Several studies have examined the role of warm-up routines in injury prevention. A systematic review by Ding et al. [29] reported a significant reduction in injury rates following structured warm-up programs in youth sports. However, the included studies primarily addressed lower-limb injuries in team sports among adolescents, with limited applicability to adult, bodyweight-based, or wrist-loaded disciplines such as handstands.

Another review by Fradkin et al. [30], which focused on randomized controlled trials across multiple sports, found limited and inconsistent evidence that warm-up routines effectively prevent injuries—highlighting the heterogeneity in interventions and injury definitions as key limitations. McCrary et al. [31], in a review specifically targeting upper body warm-ups, concluded that while certain warm-up modalities may enhance performance, no studies to date have evaluated injury prevention outcomes for the upper limb—particularly not for wrist-dominant disciplines such as calisthenics, acrobatics, or hand balancing.

In the present study, warm-up routines, though widely practiced, were not associated with a lower prevalence of chronic wrist pain. It is important to note that participants self-reported their individual warm-up routines, which were not standardized in terms of content, duration, or intensity. Moreover, whereas the aforementioned studies assessed confirmed injuries, our outcome measure was chronic wrist pain, which may represent an earlier condition not captured by prior musculoskeletal injury-focused research.

No significant association was found between weekly training hours and wrist pain prevalence in our cohort. This aligns with evidence from other sports showing that overuse injuries are influenced less by absolute training volume and more by factors such as intensity, abrupt increases in load, and biomechanical stress. Two systematic reviews in youth soccer and basketball players [32,33] concluded that overall training volume was not a reliable predictor of injury risk; instead, sudden spikes in workload or congested competition schedules were more influential. Likewise, studies in runners and across multiple sports [34,35] have demonstrated that training errors and rapid changes in load, rather than cumulative mileage or total volume, are the primary drivers of overuse injuries. Although these studies investigated predominantly lower-limb or whole-body injuries in field and endurance sports, their conclusions are consistent with our findings. Focusing specifically on the wrist, Forman et al. [36] reported that sustained, forceful contractions—rather than training duration—are the key contributors to overuse-related pathology.

Contrary to expectations, the use of wrist braces or grip devices was not associated with reduced rates of chronic wrist pain. Although a prior study among gymnasts found that a specially designed wrist brace reduced pain [37], differences in participant demographics, brace type, and training demands may account for the discrepancy. Specifically, the gymnast study evaluated adolescent athletes performing high-impact repetitive loading during growth. Moreover, the prior intervention targeted acute symptomatic relief during intensive training blocks, while our data reflect long-term, self-reported prevalence of chronic wrist pain.

The planche, an advanced calisthenic skill requiring extreme wrist extension and high upper-body control, imposes considerable compressive and shear forces on the wrists [38]. Surprisingly, participants who trained for the planche did not report higher rates of chronic wrist pain. This may reflect the advanced training experience and load management typically practiced by athletes capable of performing such maneuvers [38].

Younger age had a trend towards a higher likelihood of chronic wrist pain, this finding may reflect age-related differences in pain perception, as suggested by previous research [39,40].

Gender-based differences were also observed. Female participants reported more cautious and proactive approaches to managing wrist pain during training, aligning with literature suggesting that women tend to be more risk-averse and injury-conscious in athletic contexts [41,42].

Sport discipline may also influence pain prevalence. A non-significant trend toward higher wrist pain was observed among participants engaged in high-impact bodyweight disciplines such as acrobatics and capoeira compared to lower-impact practices. Although not statistically significant, this finding suggests a possible increased risk associated with higher-impact activities.

Beyond the physical implications, chronic musculoskeletal pain can have substantial psychological and social effects on athletes. Persistent pain has been shown to negatively impact athletic performance, mental health, and social engagement within sport communities [43,44]. Accordingly, chronic wrist pain may limit not only handstand performance but also broader well-being.

The wrist is a complex joint with limited tolerance for sustained hyperextension. Because of the close proximity of osseous, ligamentous, and tendinous structures, accurate diagnosis of wrist pain can be challenging [45]. The joint is inherently unstable due to its small articular contact areas, and axial loads—particularly in hyperextension—may easily exceed the stabilizing capacity of the carpal ligaments [46]. During handstand practice, forces are transmitted primarily through the radiocarpal and midcarpal joints, placing stress on the volar radiocarpal ligaments, the triangular fibrocartilage complex (TFCC), and the flexor tendons. Individuals with restricted wrist extension or, conversely, generalized ligamentous laxity are therefore at increased risk of pain and overuse injury [47,48].

Anatomical variations may further modify risk and predispose athletes to pain or injury during hand-stand practice. For instance, ulnar variance, defined as the relative length of the ulna compared to the radius at the wrist. Positive ulnar variance increases load transfer to the TFCC and has been strongly associated with TFCC tears and ulnar impaction syndrome [49,50]. In contrast, negative ulnar variance reduces TFCC loading but increases tension on the scapholunate ligament and is linked to Kienböck’s disease [51,52]. These variations mean that two athletes performing the same maneuver may develop very different injury profiles depending on their underlying carpal alignment.

Taken together, these findings suggest that chronic wrist pain is a common concern among handstand practitioners, with multifactorial and often complex underlying causes.

This study has several limitations. First, all data were self-reported, which introduces the potential for recall and reporting bias. Recruitment through online platforms may have further contributed to selection bias. In addition, exclusion of trauma history was based solely on participant interpretation, raising the possibility of misclassification. Importantly, no clinical examination or imaging was performed; therefore, the exact anatomical sources of pain could not be identified. This limitation prevents differentiation between intra-articular, tendinous, or TFCC-related pathology. To our knowledge, no prior study has specifically examined the pathophysiology of chronic wrist pain in handstand practitioners. Future investigations should incorporate objective diagnostic tools such as physical examination, ultrasound, or MRI, along with longitudinal follow-up, to better characterize injury mechanisms and to inform targeted preventive and rehabilitative strategies. Accurate diagnosis is not only essential for future research but also for tailoring individualized rehabilitation, treatment, and preventive strategies for athletes. Establishing diagnostic clarity will be critical in translating these findings into actionable interventions for both practitioners and clinicians.

## 5. Conclusions

Chronic wrist pain was reported by more than half of the adult handstand practitioners surveyed, confirming that it is a highly prevalent problem in this population. Younger athletes demonstrated significantly higher rates of pain, whereas training hours, warm-up routines, brace use, and grip devices were not associated with pain prevalence. These findings indicate that chronic wrist pain in handstand practice is not primarily determined by training volume or preventive routines, but rather likely reflects multifactorial influences including age, discipline type, and individual biomechanics.

This study highlights the burden of chronic wrist pain among adult handstand practitioners. The results provide an evidence base for clinicians and coaches, indicating that standard preventive strategies may be insufficient and that accurate diagnosis of underlying pathology is essential for guiding individualized treatment and rehabilitation.

## Figures and Tables

**Figure 1 jfmk-10-00372-f001:**
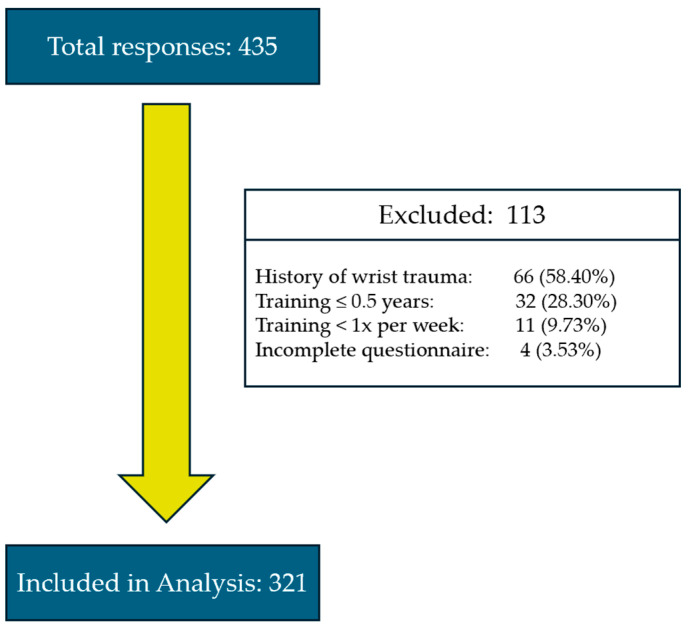
Flow diagram illustrating total respondent numbers, exclusion criteria, and final sample included in the statistical analysis.

**Table 1 jfmk-10-00372-t001:** Survey Questionnaire Used for Data Collection.

Question	Response Options
Age Group	18–25
	26–35
	36–45
	46+
Gender	Male
	Female
	Other
Main Sport	Handstand only
	Yoga
	Calisthenics
	Capoeira
	Breakdance
	Circus
	Movement
	CrossFit
	Acrobatics
How long have you been practicing handstands?	Less than 6 months
	6 months–up to 2 years
	2–up to 5 years
	5–up to 10 years
	More than 10 years
How often do you practice handstands?	Every day
	5 times a week
	3 times a week
	Once a week
	Less than once a week
Do you warm up before practicing handstands?	Yes—Wrist and finger stretching
	Yes—Wrist circumduction
	Yes—Wrist and finger stretching, circumduction, and wrist flexion
	Yes—Full-body warm-up
	No
Do you wear wrist braces while performing handstands?	Yes
	No
Do you use grip devices (e.g., parallettes, blocks)?	Yes—Always
	Yes—More than half the time
	Yes—Less than half the time
	No
Do you also practice the planche, press-ups, or other movements that involve wrist hyperextension under load?	Yes
	No
Have you ever sustained a wrist injury or undergone wrist surgery?	Yes—Distal radius fracture
	Yes—Other
	No
Have you ever experienced wrist pain due to handstand practice?	Never
	Yes—Once or twice in the past
	Yes—Recurring pain when training beyond my usual routine
	Yes—Recurring pain even during my usual routine
	Yes—Persistent pain
When does the pain typically begin?	During handstand practice
	Within 2 h after practice
	2–24 h after practice
	More than 24 h after practice
	Constant
How do you usually manage the pain?	Rest for a few days
	Use of analgesics or ice packs
	Rest combined with analgesics or ice packs
	Ignore the pain and continue training as usual

**Table 2 jfmk-10-00372-t002:** Demographic Characteristics of the Study Participants.

Variable	Category	N (%)
Gender	Male	174 (54.2%)
	Female	147 (45.8%)
Age group	18–25	69 (21.4%)
	26–35	162 (50.4%)
	36–45	61 (19.0%)
	46+	29 (9.0%)
Training experience	Half a year–up to 2 years	100 (31.1%)
	2 years–up to 5 years	102 (31.7%)
	5 years–up to 10 years	50 (15.5%)
	More than 10 years	69 (21.4%)
Training frequency	Once a week	34 (10.5%)
	3 times a week	171 (53.2%)
	5 times a week	68 (21.2%)
	Daily	48 (14.5%)
Sport discipline	Acrobatics	13 (4.0%)
	Calisthenics	39 (12.1%)
	Capoeira	60 (18.6%)
	Circus	15 (4.6%)
	CrossFit	30 (9.3%)
	Movement	52 (16.1%)
	Yoga	88 (27.4%)
	Handstand only	24 (7.4%)
Total participants		321 (100%)

Data are presented as number of participants and percentage of the total sample, expressed as N (%).

**Table 3 jfmk-10-00372-t003:** Prevalence of Chronic Wrist Pain by categorical nominal variables.

Nominal Variable	Chronic Pain N (%)	No Pain N (%)	Total	*p*-Value *
Sport Discipline				
Calisthenics	21 (53.8%)	18 (46.2%)	39	0.739
Yoga	47 (53.4%)	41 (46.6%)	88	
Acrobatics	9 (69.2%)	4 (30.8%)	13	
Movement	30 (57.7%)	22 (42.3%)	52	
Capoeira	38 (63.3%)	22 (36.7%)	60	
CrossFit	16 (53.3%)	14 (46.7%)	30	
Circus	10 (66.7%)	5 (33.3%)	15	
Handstand only	11 (45.8%)	13 (54.2%)	24	
Gender				
Female	86 (58.5%)	61 (41.5%)	147	0.626
Male	96 (55.2%)	78 (44.8%)	174	
Warm-up				
No	14 (43.8%)	18 (56.2%)	32	0.171
Yes	168 (58.1%)	121 (41.9%)	289	
Wrist brace use				
No	166 (55.9%)	131 (44.1%)	297	0.418
Yes	16 (66.7%)	8 (33.3%)	24	
Grip Devices use				
No	45 (56.3%)	35 (43.8%)	80	1.000
Yes	137 (56.8%)	104 (43.2%)	241	
Wrist hyperextension exercises				
No	96 (55.8%)	76 (44.2%)	172	0.818
Yes	86 (57.7%)	63 (42.3%)	149	

Data are presented as number of participants and percentage of the total sample, expressed as N (%). * Pearson’s Chi-square test of independence was used for all nominal categorical variables, except for Grip Devices usage, where Fisher’s exact test was applied. A *p*-value < 0.05 was considered statistically significant.

**Table 4 jfmk-10-00372-t004:** Prevalence of Chronic Wrist Pain by categorical ordinal variables.

Ordinal Variable	Chronic Pain Group N (%)	No Pain Group N (%)	Total	*p*-Value *
	Age Group			
18–25	44 (63.8%)	25 (36.2%)	69	0.223
26–35	93 (57.4%)	69 (42.6%)	162	
36–45	33 (54.1%)	28 (45.9%)	61	
≥46	12 (41.4%)	17 (58.6%)	29	
	Training Experience			
0.5–2 years	55 (55.0%),	45 (45.0%)	100	0.758
2–5 years	57 (55.9%)	45 (44.1%)	102	
5–10 years	27 (54.0%)	23 (46.0%)	50	
≥ 10 years	43 (62.3%)	26 (37.7%)	69	
	Training Frequency			
7/week	26 (54.2%)	22 (45.8%)	48	0.455
5/week	44 (64.7%)	24 (35.3%)	68	
3/week	95 (55.6%)	76 (44.4%)	171	
1/ week	17 (50.0%)	17 (50.0%)	34	

Data are presented as number of participants and percentage of the total sample, expressed as N (%). * Chi-Square test for trend was used. A *p*-value < 0.05 was considered statistically significant.

**Table 5 jfmk-10-00372-t005:** Prevalence of Chronic Pain by Warm-Up Type.

Warm-Up Type	Chronic Pain N (%)	No Chronic Pain N (%)	Total	*p*-Value *
Circumduction only	33 (55.9%)	26 (44.1%)	59	0.144
Full-body warm-up	26 (48.1%)	28 (51.9%)	54	
No warm-up	14 (43.8%)	18 (56.2%)	32	
Stretching + Circumduction	87 (64.0%)	49 (36.0%)	136	
Stretching only	21 (53.8%)	18 (46.2%)	39	

Data are presented as number of participants and percentage of the total sample, expressed as n (%). * Pearson’s Chi-square test of independence was used to assess the association between warm-up type and the presence of chronic wrist pain. A *p*-value < 0.05 was considered statistically significant.

**Table 6 jfmk-10-00372-t006:** Pain management by gender and by age group.

Variable	Rest + Analgesics N (%)	Rest N (%)	Analgesics N (%)	Ignore Pain, Continue Training N (%)	Total ^†^	*p*-Value *
Gender						
Female	19 (14.0%)	52 (38.2%)	8 (5.9%)	57 (41.9%)	136	0.016
Male	14 (8.3%)	87 (51.5%)	2 (1.2%)	66 (39.1%)	169	
Age Group						
18–25	9 (13.6%)	30 (45.5%)	2 (3.0%)	25 (37.9%)	66	0.520
26–35	12 (7.6%)	75 (47.5%)	5 (3.2%)	66 (41.8%)	158	
36–45	10 (18.5%)	21 (38.9%)	3 (5.6%)	20 (37.0%)	54	
46+	2 (7.4%)	13 (48.1%)	0 (0.0%)	12 (44.4%)	27	

Data are presented as number of participants and percentage of the total sample, expressed as N (%). * Pearson’s Chi-square test of independence was used. A *p*-value < 0.05 was considered statistically significant. † Note: Totals do not reflect the full sample size, as some participants without chronic wrist pain did not respond to pain management questions, deeming them irrelevant.

## Data Availability

The datasets generated and analyzed during the current study are available from the corresponding author on reasonable request.

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
