# Peer review of "The Wrist as a Weightbearing Joint in Adult Handstand Practitioners: A Cross-Sectional Survey of Chronic Pain and Training-Related Factors"

_jfmk, 2025, doi:10.3390/jfmk10040372_

Round 1
Reviewer 1 Report
Comments and Suggestions for Authors
The manuscript entitled “The Wrist as a Weightbearing Joint: A Cross-sectional Study in Handstand Practitioners” was reviewed. The article provides interesting information on the topic; however, adjustments need to be made so that the article can continue the path to publication.
I kindly ask if all changes made to the text be highlighted in yellow or a different color in the text.
Below are the reviewer's considerations to be adjusted in the manuscript.
Abstract:
1- Present a specific topic for the study objectives. Additionally, please provide more information in the title to support what will be covered in the study, aligning with the objectives.
2- Present the participants' sociodemographic information in the methodology. It is also important to adjust the methodology, since the chi-square test does not perform comparisons, but rather associations. What other nonparametric parameters were evaluated?
3- Did you use a specific questionnaire to assess the presence and intensity of pain? If so, please provide this information.
4- In the results, present the sociodemographic information of the sample evaluated. Present the data in absolute and relative terms so that the reader can better understand your data.
5- You are not completing the study objectives, as no biomechanical assessments were performed, only pain. Therefore, please be direct in answering the proposed objectives.
Introduction:
6- Present a paragraph in the introduction addressing the anatomical aspects of the main muscle groups and joints involved in each exercise. Based on this information, please indicate which are most affected by practitioners of the sports.
7- Present epidemiological information about these injuries/pains and how they could be prevented/treated.
8- Clarify the current state of the art on the topic and how this study could provide information that would contribute to advancement in this field.
9- The objectives at the end of the introduction are different from those presented in the objectives. Take the opportunity to review what was suggested and adapt both to be consistent.
Materials and Methods:
10- In addition to presenting the type of study, it is also necessary to mention the location (city/country) of the intervention.
11- Did you perform a sample size calculation to determine whether the minimum number of participants was recruited?
12- Please prepare a flowchart so readers can see how many people completed the questionnaire, how many were excluded, and how many remained for statistical analysis.
13- Why didn't you use a validated questionnaire to gather information about pain, since it was one of the main points of the research?
14- Regarding the description of the statistics, you stated in the abstract that you performed nonparametric analyses. However, here you present the descriptive data as if they were parametric data, that is, mean and standard deviation. The correct interpretation in this case would be median and interquartile range or minimum and maximum. I also couldn't find which normality tests were performed. I believe you should rewrite the analyses and, consequently, adjust the presentation of the data in the results. If the comparison test is used, it should be a non-parametric similar test.
15- Present ethical approval for the study in methodology. Did the participants sign the informed consent form before answering the questionnaires?
Results:
16- Please begin the results with a table detailing the participants' sociodemographic information.
17- Table 2 and the initial results information can be replaced by creating the suggested flowchart.
18- Tables 3 and 4 can be merged, and you should create a legend presenting statistical information and the established p-value.
19- Table 5 could be converted into a figure. However, the data presentation issues mentioned in the statistics should be reviewed.
20- Tables 6 to 9 can be merged, or some of them could be presented in text, as the current format makes the manuscript too cumbersome for readers to interpret. A legend is also necessary.
Discussion:
21- You started the discussion by presenting results. Please adjust the discussion to just discuss and avoid repeating information already mentioned.
22- The discussion is the time for you to further detail the articles in the literature so that the reader can understand the similarity of the findings with those of this study. Please provide more detail on the cited articles regarding methodology, participants, and types of injury/pain, and use this to explain the findings of this study.
23- Go into anatomical issues and explain what could be affecting the individuals. Also, present possible solutions so that people can avoid injuries or treat them effectively.
24- Review the limitations of the study, based on what was questioned throughout the evaluation.
25- Present possible practical implications for your study.
Conclusions:
26- Please elaborate on a more assertive conclusion regarding the research objectives.
27- Appendix A should be removed from the main work and included in the supplementary materials.
28- The references are somewhat outdated; please replace the old ones with publications from the last three years, that is, from 2022 onward.
Author Response
Abstract:
Comment (title) : please provide more information in the title to support what will be covered in the study, aligning with the objectives.
Response (title):
We have updated the title from:
"The Wrist as a Weightbearing Joint: A Cross-sectional Study in Handstand Practitioners"
to:
"Chronic Wrist Pain and Associated Risk Factors Among Handstand Practitioners: A Cross-Sectional Survey Study"
[page 1, lines 2-4]
Comment 1: Present a specific topic for the study objectives.
Response 1:
we have revised the final sentence of the Introduction part of the Abstract from:
“This study aimed to examine the role of the wrist as a weightbearing joint in handstand practitioners.”
to the following:
“This study aimed to investigate the prevalence of chronic wrist pain and to explore associated factors such as discipline, training habits, and pain management strategies.”
[page 1, lines 14-16]
Comment 2: Present the participants' sociodemographic information in the methodology. It is also important to adjust the methodology, since the chi-square test does not perform comparisons, but rather associations. What other nonparametric parameters were evaluated?
Response 2:
- we clarified in the manuscript that participants were aged 18 years or older, of any gender, and actively engaged in handstand practice. More detailed inclusion and exclusion criteria are now explicitly provided in the full Methods section. [page 1, lines 18-20].
- In the abstract, we now describe that associations (rather than comparisons) were evaluated using appropriate tests for categorical data. [page 1, lines 28-30]. In the Methods section, we state that Fisher’s exact test and Pearson’s Chi-square test were used for nominal variables, while the Chi-square test for trend was used for ordinal variables. No other nonparametric tests (e.g., Mann–Whitney U or Kruskal–Wallis) were used, as all variables were categorical rather than continuous.
Comment 3: Did you use a specific questionnaire to assess the presence and intensity of pain? If so, please provide this information.
Response 3: As noted in the full Methods section, the questionnaire was specifically designed for this study and has not been previously validated or published [page 3, lines 152-154]. To improve clarity, we have now also added the following sentence to the Abstract: “via a self-developed questionnaire designed for this study.”
[page1, lines 26-27]
Comment 4: In the results, present the sociodemographic information of the sample evaluated. Present the data in absolute and relative terms so that the reader can better understand your data.
Response 4: We added a sentence in the results:
A total of 321 participants were included in the study. The most represented age group was 25–34 years, comprising 123 (38.3%) of the participants. Gender distribution was 174 (54.2%) males and 147 (45.8%) females. The most common sport disciplines were Yoga (88, 27.4%), Capoeira (60, 18.7%), and Movement (52, 16.2%).
[page 1, lines 32-35]
Comment 5: You are not completing the study objectives, as no biomechanical assessments were performed, only pain. Therefore, please be direct in answering the proposed objectives.
Response 5: We thank the reviewer for this observation. We acknowledge that the original phrasing of the objectives may have implied a broader biomechanical investigation. The primary focus of the current study was on pain patterns, training characteristics, and associated factors in handstand practitioners, rather than on direct biomechanical assessments.
[page 1 , lines 14-16]
Introduction:
Comment 6: Present a paragraph in the introduction addressing the anatomical aspects of the main muscle groups and joints involved in each exercise. Based on this information, please indicate which are most affected by practitioners of the sports.
Response 6: We have now added a paragraph to the Introduction section discussing the biomechanics of the wrist and upper limb, including the key agonists, stabilizers, and synergists involved in wrist movement and stability [page 2, lines 55-66].
Comment 7: Present epidemiological information about these injuries/pains and how they could be prevented/treated.
Response 7: As noted in the manuscript, most available epidemiological data on wrist injuries in weight-bearing activities comes from studies in gymnasts. We have now expanded this section to include more detailed epidemiological data. [page 2, lines 95-101].
To address the second part of the comment regarding prevention and treatment, we have now included a summary of commonly proposed preventative strategies [page 2, lines 105-107]
Comment 8: Clarify the current state of the art on the topic and how this study could provide information that would contribute to advancement in this field.
Response 8:
The revised manuscript now presents a broader spectrum of handstand-practicing disciplines [page 2, lines 81-90].
Additionally, we now emphasize that most prior research has focused on youth or elite gymnasts—a population exposed to high-volume and high-impact training from a young age. However, these findings may not be applicable to the growing number of recreational or adult athletes practicing handstand-based disciplines, which differ in training frequency, intensity, and biomechanical demands [page 3, lines 101-102].
Comment 9- The objectives at the end of the introduction are different from those presented in the objectives. Take the opportunity to review what was suggested and adapt both to be consistent.
Response 9:
As noted in our response to Comment 5, we have revised the Objectives in both the Abstract and the Introduction to ensure consistency and alignment with the actual scope of the study. Specifically, we clarified that the aim of this study was to characterize the occurrence of wrist pain among adult handstand practitioners and to explore associated variables such as sport discipline, training volume, and self-reported coping strategies—rather than conducting biomechanical assessments. These updates have been made in both sections and highlighted in the revised manuscript [page 3, lines 110-111].
Materials and Methods:
Comment 10: In addition to presenting the type of study, it is also necessary to mention the location (city/country) of the intervention.
Response 10: We have now clarified that the study was conducted in Israel, with recruitment through Hebrew-language social media platforms and gym communities. The original questionnaire was administered in Hebrew, and only translated into English after data collection was completed. While participation was technically open to all, all respondents were ultimately Israeli due to the language and recruitment approach. These details have now been added to the Methods section [page 3, lines 114-122]
Comment 11: Did you perform a sample size calculation to determine whether the minimum number of participants was recruited?
Response 11: As this was an exploratory, cross-sectional survey aimed at characterizing a novel population (adult handstand practitioners), no formal a priori sample size calculation was performed. Instead, we sought to maximize participation during the data collection period by distributing the survey broadly via online handstand communities. We have now added this clarification in the methods section:
No formal sample size calculation was performed due to the exploratory nature of the study; participation was maximized through open online recruitment over a defined time period
[page 5, lines 145-147]
Comment 12: Please prepare a flowchart so readers can see how many people completed the questionnaire, how many were excluded, and how many remained for statistical analysis.
Response 12: we have now created and included a participant flowchart (Figure 1) to illustrate the total number of respondents, any exclusions, and the final analytic sample.
[page 6, line 156]
Comment 13: Why didn't you use a validated questionnaire to gather information about pain, since it was one of the main points of the research?
Response 13: While validated tools such as the Patient-Rated Wrist Evaluation (PRWE) and the Disabilities of the Arm, Shoulder, and Hand (DASH) questionnaires are widely used to assess wrist pain and upper limb dysfunction, they primarily evaluate limitations in daily activities and general functional tasks.
Given that our target population comprises adult handstand practitioners—many of whom are healthy, athletic individuals engaging in advanced physical activity—we required a tool that would specifically capture handstand-related wrist pain, training variables, and coping strategies relevant to this unique context.
Comment 14:
Regarding the description of the statistics, you stated in the abstract that you performed nonparametric analyses. However, here you present the descriptive data as if they were parametric data, that is, mean and standard deviation. The correct interpretation in this case would be median and interquartile range or minimum and maximum. I also couldn't find which normality tests were performed. I believe you should rewrite the analyses and, consequently, adjust the presentation of the data in the results. If the comparison test is used, it should be a non-parametric similar test.
Response 14: We thank the reviewer for this important observation. In the original version, we mistakenly reported mean ± SD while also describing nonparametric testing, which created inconsistency. After reviewing the dataset, we clarified that all survey variables were categorical and therefore treated as such. In the revised manuscript, descriptive data are now presented as absolute counts and percentages. Comparative analyses were conducted using Pearson’s Chi-square or Fisher’s exact test, as appropriate. No parametric analyses were performed, and measures of central tendency such as mean ± SD are no longer reported. We have also corrected the Statistical Analysis section and Results to reflect this approach.
[page 5, lines 139-144]
Comment 15: Present ethical approval for the study in methodology. Did the participants sign the informed consent form before answering the questionnaires?
Response 15: In accordance with the MDPI submission format, the ethical approval and consent statement were originally placed at the end of the manuscript. As requested, we have now also included this information in the Materials and Methods section. An explanation of the study and an informed consent form were presented online as a mandatory step before completing the survey, and participants provided consent electronically. A copy of the consent form was submitted to the editor at the time of the original submission.
[page 3, lines 118-119]
Results:
Comment 16: Please begin the results with a table detailing the participants' sociodemographic information.
Response 16: We have added a table at the beginning of the Results section detailing the participants' sociodemographic characteristics (Table 2). As this information now appears comprehensively in the table, we have removed the previously included figure displaying sport discipline distribution to avoid redundancy [page 6, line 162].
Comment 17: Table 2 and the initial results information can be replaced by creating the suggested flowchart.
Response 17: As requested, we removed Table 2 and replaced it with a flowchart (Figure 1) illustrating participant recruitment and exclusion.
Comment 18: Tables 3 and 4 can be merged, and you should create a legend presenting statistical information and the established p-value.
Response 18: We have merged Tables 3 and 4 into a single table of nominal categorical variables, as suggested. We also added a detailed legend specifying the statistical tests used for each variable and a p-value threshold of < 0.05 for significance [page 7, line 178-180].
Comment 19: Table 5 could be converted into a figure. However, the data presentation issues mentioned in the statistics should be reviewed.
Response 19: Thank you for this suggestion. While we opted to retain a table format for clarity and consistency, we have now created a unified table of ordinal categorical data (Table 4). In addition, we reviewed and corrected the data presentation issues: numerical variables were removed, and all results are now presented as frequencies and percentages, in line with their categorical nature.
[page 8, lines 182-183]
Comment 20: Tables 6 to 9 can be merged, or some of them could be presented in text, as the current format makes the manuscript too cumbersome for readers to interpret. A legend is also necessary.
Response: In accordance with your comment, we have revised the manuscript by consolidating the data:
All nominal variables are now presented together in a single table (table 3). All ordinal variables are grouped into a separate table. Each table now includes a clear legend specifying the statistical tests used.
The table addressing warm-up habits was intentionally kept separate, as we believe it is important to highlight this specific finding. Although warm-up is widely considered protective, our data did not demonstrate a significant association with reduced pain. For this reason, we chose to emphasize it distinctly (Table 5) [page 8, lines 185-186]
.
Discussion:
Comment 21: You started the discussion by presenting results. Please adjust the discussion to just discuss and avoid repeating information already mentioned.
Response 21: We revised the opening of the Discussion section to reduce repetition and focus more on interpretation and comparison with existing literature.
[page 9, lines 205-210]
Comment 22: The discussion is the time for you to further detail the articles in the literature so that the reader can understand the similarity of the findings with those of this study. Please provide more detail on the cited articles regarding methodology, participants, and types of injury/pain, and use this to explain the findings of this study.
Response 22: We expanded the Discussion to provide more detail on the methodology, study populations, and types of injuries reported in the cited literature. Specifically, we added comparisons regarding warm-up practices, training volume, and the use of grip devices, highlighting how these factors were studied in different athletic populations.
[page 9, lines 212-243]
Comment 23: Go into anatomical issues and explain what could be affecting the individuals. Also, present possible solutions so that people can avoid injuries or treat them effectively.
Response 23: We expanded the Discussion to describe anatomical factors that predispose individuals to wrist pain during handstand practice, including ulnar variance. Instead of proposing generic preventive strategies, we highlighted the importance of accurate diagnosis in both research and clinical settings. Identifying the specific anatomical source of pain is essential for tailoring appropriate rehabilitation, treatment, and prevention programs.
[page 10, lines 261-275 ]
Comment 24: Review the limitations of the study, based on what was questioned throughout the evaluation.
Response 24: A limitations section was already included, but we further emphasized the most critical issue: the reliance on self-reported pain. Because no clinical examination or imaging was performed, the only available variable was the presence or absence of pain, without insight into its underlying etiology. We also acknowledged the risks of recall bias, selection bias, and potential misclassification of trauma history.
[page 11, lines 276-282]
Comment 25: Present possible practical implications for your study.
Response 25: We added a statement on the practical implications of our findings. Specifically, we note that accurate diagnosis is not only essential for future research but also for tailoring individualized rehabilitation, treatment, and preventive strategies for athletes. Establishing diagnostic clarity will be critical in translating these findings into actionable interventions for both practitioners and clinicians.
[page 11, lines 282-286]
Conclusions:
Comment 26: Please elaborate on a more assertive conclusion regarding the research objectives.
Response 26: The conclusion section was revised to be more assertive and results-focused. It now explicitly states the prevalence of chronic wrist pain, the higher risk among younger athletes, and the lack of association with training hours, warm-up routines, or brace/grip device use.
[page 11, lines 288-296]
Comment 27: Appendix A should be removed from the main work and included in the supplementary materials.
Response 27: Appendix A has been removed from the main manuscript as requested.
Comment 28: The references are somewhat outdated; please replace the old ones with publications from the last three years, that is, from 2022 onward.
Response 28: The reference list has been updated accordingly. In the revised version, the majority of the newly added references are from 2023 and 2024.
Reviewer 2 Report
Comments and Suggestions for Authors
The article submitted for analysis is a relevant study that reveals the issue of wrist pain during implementation of systematic handstand practices. Particular relevance of this study is emphasized by the author's attention to such forms of physical exercise, which are mainly of a health or recreational nature. The study of impact and frequency of pain factors can become the basis for substantiating principles of prevention and reduction of injury risks.
The article is structured with a consistent logical presentation of information. The purpose of study corresponds to its topic. The author conducted an analysis of reference sources, taking into account research papers over the past 5 years.
The article is well illustrated, the use of tables and figures simplifies perception of information.
I cannot but agree with the author regarding limitations of this study, and the need for further expansion of this study based on widespread use of instrumental research methods.
Unfortunately, during review process, issues arose that require additional attention:
- In the introduction and further in discussion, authors should have paid more attention to specific examples of exercises that involve intensive load on the wrist.
- Theoretical substantiation of the article topic can be expanded by considering this scientific publication:
Keren T. A Distal Radial Metaphyseal Stress Fracture in an 11-Year-Old Capoeira Dancer: A Case Report. Journal of Dance Medicine & Science. 2016;20(4):181-183. doi:10.12678/1089-313X.20.4.181 https://www.scopus.com/pages/publications/85164188758
- It should be clarified which version of the questionnaire the participants were dealing with (Hebrew, English)?
- The questionnaire (see Table 1) does not include a group of questions regarding pain assessment.
- In our opinion, a description of the study contingent should be presented in the Materials and methods section, which already partially presents information on formation of study participants group.
- Authors should provide extended information regarding the age of study participants and explain exactly how many participants in each age group were analyzed (e.g., line 135 - age group was 25–34 years, comprising 123 (38.3%) of the participants, and in Table 9 the number of participants in the age group 25–34 years -158 people).
- Please, agree on the names of tables 7, 8, 10 and their contents.
- Additionally analyze the content of Table 5 and agree on the possibility of presenting data in form of mean value and standard deviation (Mean ± SD).
- In lines 196 – 199 we consider it inappropriate to analyze and draw conclusions based on the results of participants who met the exclusion criteria.
- The conclusions need to be corrected in accordance with the need to better reflect results of the study and assess the degree to which its goal has been achieved, rather than expressing assumptions and considering prospects for future research.
Author Response
Comment 1. In the introduction and further in discussion, authors should have paid more attention to specific examples of exercises that involve intensive load on the wrist.
Response: We thank the reviewer for this important suggestion. In response, we have added a detailed biomechanical overview of wrist-loading exercises commonly performed in handstand practice and related disciplines, including press-to-handstand, handstand walking, and acrobatic skills such as round-offs and cartwheels. This content was added to the Introduction section, following the paragraph on wrist muscle engagement [page 2, lines 64-73].
Comment 2. Theoretical substantiation of the article topic can be expanded by considering this scientific publication:
Keren T. A Distal Radial Metaphyseal Stress Fracture in an 11-Year-Old Capoeira Dancer: A Case Report. Journal of Dance Medicine & Science. 2016;20(4):181-183. doi:10.12678/1089- 313X.20.4.181 https://www.scopus.com/pages/publications/85164188758
Response:
We have incorporated the suggested reference by Keren et al. into the revised manuscript and expanded the theoretical background by providing a more detailed overview of wrist-related injuries across multiple wrist-loading disciplines, including Capoeira, breakdancing, yoga, circus arts, and CrossFit. The case report by Keren et al., describing a distal radial metaphyseal stress fracture in a young Capoeira practitioner, is now specifically cited to illustrate the potential for overuse injuries associated with hand-supported acrobatic movements [page 2, lines 81-90].
Comment 3: It should be clarified which version of the questionnaire the participants were dealing with
(Hebrew, English)?
Response 3: We have now clarified that the study was conducted in Israel, with recruitment through Hebrew-language social media platforms and gym communities. The original questionnaire was administered in Hebrew, and only translated into English after data collection was completed. While participation was technically open to all, all respondents were ultimately Israeli due to the language and recruitment approach. These details have now been added to the Methods section [page 3, lines 120-122].
Comment 4. The questionnaire (see Table 1) does not include a group of questions regarding pain
assessment.
Response 4: The full questionnaire, including the pain assessment items, has now been added to Table 1 in the revised manuscript.
[page 4, line 137 ]
Comment 5: In our opinion, a description of the study contingent should be presented in the Materials and methods section, which already partially presents information on formation of study participants group.
Response 5: We thank the reviewer for this comment. The study cohort description is presented at the very beginning of the Results section, in accordance with reporting conventions. In addition, a new demographics table (Table 2) has been included [page 6, line 162].
Comment 6: Authors should provide extended information regarding the age of study participants and explain exactly how many participants in each age group were analyzed (e.g., line 135 - age group was 25–34 years, comprising 123 (38.3%) of the participants, and in Table 9 the number of participants in the age group 25–34 years -158 people).
Response 6: We appreciate the reviewer’s careful reading. The discrepancy arose from a clerical aggregation/transcription error in the text and not from any change to our dataset or age-grouping scheme. The underlying dataset and participant assignments were constant throughout; age was categorized as 18–25, 26–35, 36–45, and 46+ for all analyses. We have corrected the counts in the text so they now match the verified frequencies presented in the revised demographics table: 18–25: 69 (21.4%); 26–35: 162 (50.4%); 36–45: 61 (19.0%); 46+: 29 (9.0%). No analyses or conclusions are affected by this correction [page 5, line 152, page 6, line 162].
Comment 7: Please, agree on the names of tables 7, 8, 10 and their contents.
Response 7: In response to another reviewer’s request to reduce the number of tables, several tables were merged and renumbered. The revised version now includes consolidated tables with harmonized titles and contents. We hope that the revisions resolve the concerns raised.
Comment 8: Additionally analyze the content of Table 5 and agree on the possibility of presenting data in form of mean value and standard deviation (Mean ± SD).
Response 8: We have now created a unified table of ordinal categorical data. In addition, we reviewed and corrected the data presentation issues: numerical variables were removed, and all results are now presented as frequencies and percentages, in line with their categorical nature [Tale 4: page 8, lines 183-185].
Comment 9. In lines 196 – 199 we consider it inappropriate to analyze and draw conclusions based on the results of participants who met the exclusion criteria.
Response: The section referring to participants who met the exclusion criteria has been removed from the revised manuscript.
Comment 10: The conclusions need to be corrected in accordance with the need to better reflect results of the study and assess the degree to which its goal has been achieved, rather than expressing assumptions and considering prospects for future research.
Response 10: The conclusion has been corrected to better reflect the actual results rather than focusing on assumptions or future directions. We emphasized that the study achieved its primary goal of documenting the prevalence of chronic wrist pain and analyzing potential associated factors in handstand practitioners.
[page 9, lines 288-296]
Round 2
Reviewer 1 Report
Comments and Suggestions for Authors
Dear Authors,
Thank you for providing the revised version of the manuscript. After carefully reviewing the changes to the study, I was able to verify that the reviewer's requests were met. Therefore, I suggest that the manuscript be accepted for publication.